# Clinical Care Pharmacists in Urgent Care in North East England: A Qualitative Study of Experiences after Implementation

**DOI:** 10.3390/pharmacy7030114

**Published:** 2019-08-09

**Authors:** Jody Nichols, Rosie England, Stuart Holliday, Julia L Newton

**Affiliations:** 1Academic Health Science Network North East North Cumbria, Biomedical Research Building, Campus for Ageing & Vitality, Newcastle upon Tyne, River Tyne, Newcastle NE15 8NY, UK; 2North East Ambulance Service NHS Foundation Trust, Bernicia House, Goldcrest Way, Newburn Riverside, Newcastle upon Tyne, Newcastle NE15 8NY, UK

**Keywords:** integrated urgent care pharmacists (IUCCPs), pharmacists, qualitative, clinical, urgent care, emergency care, Academic Health Science Network (AHSN)

## Abstract

Our objective was to explore the implementation of a novel NHS England (NHSE)-funded pilot project aimed at deploying clinical pharmacists in an integrated urgent care (IUC) setting including the NHS 111 service. Eight integrated urgent care clinical pharmacists (IUCCPs) within the participating North East of England Trusts. Individuals participated in semi-structured 1-to-1 interviews by an experienced qualitative researcher, either face-to-face or via the telephone. Each recording was transcribed, and the five stages of framework analysis (familiarisation, identifying a thematic framework, indexing, charting, mapping and interpretation) took place to establish emerging themes. All interviews took place between November 2018–February 2019. Four higher-order themes were identified: 1. Personality Traits, 2. Integration, 3. Benefits, 4. Training. The IUCCP programme is an innovative NHSE initiative. It provides an opportunity to extend the role of clinical pharmacists into the hard-pressed clinical environment of urgent and emergency care. Our evaluation has highlighted the potential for this professional group to contribute clinically in this area. Better communications, standard operating procedures and induction will improve how individuals develop in these novel roles.

## 1. Introduction

The vast majority of pharmacists in the UK are currently based in secondary or community care, but in order to support the transformation outlined in the Five-Year Forward View [1], and to deliver the aspirations of the NHS Long Term Plan [2], a new Pharmacy Integration Fund was set up in October 2016. The new Pharmacy Integration Fund highlights the benefits and provides an opportunity to extend the role of clinical pharmacists into the hard-pressed clinical environment of urgent and emergency care. The initial priorities for the fund identified through consultation and working with stakeholders including commissioners, providers, regulators, professional bodies and patient groups were: (1) The deployment of clinical pharmacists and clinical pharmacy services in primary care including groups of general practices, care homes and urgent care settings such as NHS 111. (2) The development of “infrastructure” through the pharmacy professional workforce, accelerating digital integration and establishing the principles of medicines optimisation for patient-centred care. Integrated urgent care (IUC)/NHS 111/NHS England (NHSE) and the Health Education England (HEE) Workforce Development Programme has undertaken initial pilot studies to evaluate the role of the clinical pharmacist working within the NHS 111 contact centre [3,4]. This pilot work has shown that pharmacists can add value to the clinical skill mix working within the IUC Clinical Assessment Service (CAS), completing calls and providing self-care advice across a range of calls that involve the use of medicines. In addition, recent evidence [5] supports the potential for clinical pharmacist involvement in the emergency department.

The pilot scheme aimed to facilitate pharmacists working in IUC as part of the multidisciplinary team as an expert resource. This involved handling medicines-related enquiries and issues, undertaking clinical assessment and treatment of minor ailments prescribing where appropriate, prescribing for repeat prescription requests and providing self-care advice.

The North East and North Cumbria (NENC) proposal set out to implement a role for IUCCPs across the entire urgent care system. Proposals highlight that IUCCPs were recruited via individual Trusts utilising the standard NHS recruitment routes, and no additional incentives were offered to take urgent care roles. It was envisaged that by providing opportunities for individual pharmacists to work across a range of urgent care settings, they would develop a breadth of experience, skill and perspective enabling them to excel in each individual setting. The specific settings include (but are not limited to): Urgent Treatment Centres, Emergency Departments, Clinical Assessment Service, Out of Hours services and Mental Health Links into local Crisis teams. We believed that this integrated model provided a breadth of urgent care experience together with appropriate professional support delivered in a way that included face-to-face interactions with patients in addition to the telephone delivery via the 111 service.

The aim of the NENC (North East and North Cumbria) scheme was to allow pharmacists to develop an excellent perspective on the whole care system, bringing this to bear in every patient interaction they undertake.

The AHSN NENC (Academic Health Science Network North East and North Cumbria) has played a pivotal role in the co-ordination, evaluation and oversight of this initiative, bringing the group together as a community of practise aimed at sharing experiences and examples of good practise. Here we explore the integration process and experiences for pharmacists within urgent care settings with a view to informing this scheme’s development and future opportunities for engagement of clinical pharmacists going forward.

## 2. Methods

The evaluation was pursued through qualitative research methods using a grounded theory methodology, utilising a framework analysis analytical method to facilitate the analysis of data. This research has constructionist ontology and an epistemological position described as interpretivist. It is reinforced by social constructionism and is underpinned by an inductive approach [6,7,8].

Purposeful sampling methods were utilised during the evaluation; the evaluation specifically recruited from the eight IUCCPs employed on the scheme, and all IUCCPs employed were interviewed.

Semi-structured interviews were conducted with the eight IUCCPs within the participating North East of England Trusts. The interview schedule for the interviews is described in Table 1. All pharmacists were interviewed by an experienced qualitative researcher either face-to-face or by telephone. Each recording was transcribed, and the five stages of framework analysis (familiarisation, identifying a thematic framework, indexing, charting, mapping and interpretation) took place to establish emerging themes. The interviewer developed a manual-based system during analysis utilising “cut and paste” methodology. All interviews took place from November 2018 to February 2019.

The study was considered to be a service improvement project, and all pharmacists were asked to read and sign a consent form prior to the interview which permitted storage and publication of quotes directly from the transcribed interviews.

In addition, in December 2018, seven of the eight pharmacists attended a focus group where, in an unstructured unprompted manner, each pharmacist detailed their perception of their integration in order to compare with each other. Notes were made from these interactions via the AHSN staff member. These notes were then analysed using the analytical methods of framework analysis (as detailed above) in order to identify themes. Analyses of interviews and focus group were conducted by one researcher. Triangulation of the focus group with the interviews took place after the analytical process when the interviews were complete and the themes apparent. The thematic framework developed from the interviews was utilised to form the basis of the thematic framework for the focus group to allow for cross-referencing and further analysis so that new themes could form and develop.

To establish rigour and trustworthiness, a reflective diary was completed to review the researcher’s thoughts/feeling and experiences while interviewing and interpreting data.

### Results

The interviews were completed within the first 6 months of commencing with the programme. Four higher-order themes were identified (Table 2).

Key characteristics of interviewees:Age: 25–31 years of age.Number of years qualified: 4–9 years.Education level: Master’s Degree in Pharmacy (MPharm).Gender: 5 women; 3 men.

## 3. Higher Order Themes

### 3.1. Personality Traits

An interesting association was noted whilst interviewing the IUCCPs. All pharmacists were keen to develop careers within pressured and stressful environments, and actively discussed that they really enjoyed these high-pressure situations.

“I quite like being busy. I would rather have too much to do than have too much time on my hands. Which is why I always ended up with multiple wards and multiple things because generally I like to keep myself busy. The impression I had was that urgent care would be a time pressured environment. I thought I would quite enjoy the stress and pressure of that” IUCCP 1.

“I’ve always quite enjoyed that fast pace, making quick decisions and logical thinking rather than sitting round waiting for days for potential social aspects, and more slower paced, I quite like the fast pace”. IUCCP 2.

“I think it was more the fast pace and the continual unknown of emergency care, so never know what is going to be presented in front of you. It can be something very generic that you would see on a day to day basis, or something completely one in a million presentation that you only read about in textbooks and you can get a little more involved in it”. IUCCP 3.

### 3.2. Integration

It was noted that there was a lack of understanding and clarity around the new role from the start, during recruitment and integration into the emergency departments, with some pharmacists expressing much frustration.

“So in terms of the whole process the only thing that I would say is I feel like from, from our Trust, weren’t really told about the process well enough. I didn’t know that I was going into a pilot. Not really, until I went to that first meeting and I was like, oh, right, now I see. I’m doing this new thing. I knew it was a new initiative”. IUCCP 4.

“We weren’t told from our Trust properly what was going on. And there were other pharmacists out there that were changing their roles, and this was a new thing that they were doing and they were fitting into the urgent care system. We were just told that this is a new exciting thing where you are going to be working with NHS 111 and that was about it. You are going to be doing some shifts over there and it will be great experience, and something to add to your CV, and things like that”. IUCCP 4.

“So it didn’t work well at all … and basically it seemed like all the high level discussions about how great it would be to have a pharmacist in this post and agreeing the funding, it seemed like all of that had happened but the reality of what I would be doing and where I would be working and the conversations with urgent care and the emergency department, it seemed like that just didn’t happen prior to me coming into post which in my opinion it should have”. IUCCP 8.

Pharmacists reported a lack of a clearly defined role, and indeed some departments were not aware that they had been recruited; this made integration difficult.

“I think at the minute I haven’t really got a proper defined role and job tasks that I can do and contribute to. So I’ve got time on my hands which I’m not used to having”. IUCCP 1.

“I wouldn’t say gone particularly well. No. I feel like I have wasted hours and hours doing things without any real result at the end of it”. IUCCP 1.

“I met the consultant and he basically said he was annoyed that he didn’t know that I was coming and I was annoyed that he didn’t talk. We were both saying the same thing. So my understanding of the role is that we think we are going to develop but really being a traditional pharmacist in Accident and Emergency (A&E) right now because I have never done…”. IUCCP 5

“One thing that was a bit frustrating in the beginning was that because I was the first pharmacist in this role it’s not very clear, people don’t really know what I can do, what I can’t do. And they don’t really know what they need if that makes sense. So initially when I went to urgent care, they didn’t know they were getting a pharmacist. What is it you are going to do? So it was a bit difficult to start to see where it was that they wanted me”. IUCCP 6.

It should be noted that two of the developed themes overlapped—that is, integration and benefits. It was generally felt that integration into the Trust Departments would have been better had the A&E teams been aware of the benefits that they (the pharmacists) could bring.

“The main thing would be if A&E had buy in, it was something they could identify and understand the value of. That would be the biggest thing really. I feel at the minute there is nobody who. I probably don’t feel supported. Nobody necessarily wants me in there”. IUCCP 1.

“It’s so brand-new half the problem is trying to get to know people. And seeing the need. If you don’t have a clue what the need is how can you do anything”. IUCCP 5.

“And I think that wasn’t properly explained. I tried to get some meetings to have a conversation and explain and I was stonewalled from that point so I couldn’t even explain. So judgements were made about what I could and couldn’t do, and what the role was about before I had even had a chance to explain to anyone or talk to anyone about what it was and where I could potentially add value”. IUCCP 8.

It should be noted that not all pharmacists had the same experience of integration, with some pharmacists describing how their integration into the emergency department was smooth because they had advocates in the system.

“They all knew I was coming. I just turned up one day and got on with it essentially. We have been involved in the clinical governance meetings within the emergency department as well. Plans for guideline implementation. Clinical Audit. Things like that. A lot of different things going on at once but the day to day role is generally just screening of patients, identifying, some days you have quite a lot of input, some days not as much, it just depends. But there is always something to be getting on with, that I can do in the background that can be used to aid us going forward”. IUCCP 3

“As we have developed our input within the role people within the department know who we are now, we are a port of call in terms of pharmacy related support”. IUCCP 3.

“Within our second day we were introduced to everyone at a clinical governance meeting. So no integration issues at all. We do have that structure. We have a pharmacist that understands our competence and is keen to develop it. But also is quite strong in protecting us. So not having someone go—just do that—he is like—no they are not appropriate, stop. He is very good at that”. IUCCP 2.

The experience of integrating into the 111/Clinical Assessment Service (CAS) seemed to be similar for all the pharmacists interviewed. All the pharmacists seemed to feel fully integrated into the 111 system and working at a level they felt appropriate. It should be noted however that initially the pharmacists did identify a few problems, such as poor access to computer systems, heavy reliance on 111 colleagues and the lack of standard operating procedures to refer to.

“The only thing that is a bit strange at the minute is you get a certain number of calls that come through to the clinical assessment hub and if there is nothing there that I can competently tackle I am just flicking through it just to get a brief overview of what is going on, and if I can’t answer their calls it’s a bit like, what should I do. But apparently, they have a new system coming in quite soon that is going to be able to match up a little bit more of the low acuity work with the high acuity work that comes through to the hub so hopefully that will change.”. IUCCP 3.

“Within 111 it’s been a bit of a slow start. There’s been a lot of niggling things to sort out in terms of access to computer systems, a number of the resources such as summary care record, general medication information, that we are so used to using within our work, we rely on it quite heavily as well, particularly when you don’t have that face to face contact with a patient. So that has been a bit of a struggle getting that sorted” IUCCP 3.

“When we first started there was no way of them actually identifying what calls are appropriate to us. But it is something that they recognise and it was to do with computer systems and they are updating, so that should improve”. IUCCP 2.

“I think we have been viewed very positively by the team. Everyone has been very welcoming. Embraced our role. I think over and above just answering the queries I think we’ve been asked other stuff, like day to day, so like one of the paramedics is in charge of resources and what prescriptions they should have, to medicines information reference texts”. IUCCP 7.

“So it was really nice. YYY was really supportive and we met and he showed me around and then gave me to a clinician and then I sat with them for a shift. So I understood for one shift. Then the next shift I watched what the clinical … team did. And then the next shift I started answering some calls”. IUCCP 5.

It was observed in interviews that integration would have been much smoother if the induction process at the 111 service were more cohesive.

Another problem identified with integration was deskilling of the pharmacists themselves.

“So I’ve done my prescribing and I haven’t continued with the clinical examination side of things since I’ve had the qualification really and I think normally prescribers are generally nurses and they are really hot on that sort of thing. And I don’t have those skills because I haven’t maintained them. So I can’t work to the same level as what they maybe anticipated possibly”. IUCCP 3.

### 3.3. Benefits

It was noted that the pharmacists, despite the early integration problems, have seen and experienced real benefits of their roles within either UC or 111, for example, preventing the wrong type of medication being administered and consistency of care. A good example is highlighted where pharmacists prevented the wrong medicines from being administered for a haemorrhagic stroke.

“So we ended up having this guy who came in with a headache and we managed to get him a CT scan because they were just across the road and he was having a haemorrhagic stroke so obviously we were getting him ready to ambulance him and send him up to hospital and the GP was saying we’ll give him some aspirin and I was saying “no, don’t give that” because the type of stroke it was it wouldn’t be appropriate to give that medication”. IUCCP 8.

“From the point of view of the emergency department I think the bulk of the good work is from continuity of care, making sure there is no change in what they would usually expect when they are at home”. IUCCP 3.

Another good example of service improvement is when a pharmacist set up processes within urgent care to screen for acute or chronic illness, for example, Parkinson’s disease.

“The idea is for us to screen for any acute or chronic medication issues, so for example if they are a Parkinson’s patient and they use Parkinson’s meds, the idea is that we identify that early in the process so that they are not missing any of these medications that could potentially lead to a further deterioration in their care and making sure that they have been prescribed and administered in a timely fashion. A lot of the times these patients do get admitted to inpatient wards and they would get this process later in the line, but it is making sure that there isn’t any avoidable harm that could be caused to the patient through not receiving medication”. IUCCP 3.

A number of pharmacists highlighted how clinicians might have reacted when other practitioners (e.g., nurse practitioners) were new in the urgent care environment.

“Think because it is so new everyone … working in A&E and urgent care there are nurse practitioners and I think to myself there must have been a time when they started that everyone was thinking, Oh, what is the value of a nurse practitioner in this, what can they do? And now that they are integrated into the service they are just an additional part of the team. So I think because it is fairly new people haven’t started to realise the benefits of having a pharmacist yet, but once we show what we can do in terms of adding patient safety and providing clinical support to the rest of the team then it will be a worthwhile role having a pharmacist there”. IUCCP 6.

The IUCCPs also discussed other benefits in terms of access to medication within the hospital setting and dealing with complicated medicines.

“So I helped get the drug because it wasn’t available in A&E because it’s not used commonly. So I helped the team find where it was in the hospital and we didn’t even have it in pharmacy so I had to borrow if from one of the cardiology wards. And advised how to run the infusion. So that’s the kind of thing sometimes we’d do in resus”. IUCCP 2.

“I am better at dealing with medication things but I think we are better at dealing with complicated patients. So in a way the urgent treatment centre is probably not the best use of us. We probably would be better off in an ED setting where medications are more complicated because that is where our advice is needed. Whereas in the urgent treatment centre it is simple analgesia and things”. IUCCP 8.

### 3.4. Training

In their interviews, all pharmacists identified problems in the mandatory training course they were required to attend as part of their programme, with issues around content, absence of accreditation and the lack of applicable learning for this new role.

“The course—it’s not bad. I think it’s got somethings that are sort of useful and you can see the benefit in parts of it, but I think it’s quite generalist. It’s not very specific to our needs in the North East. I think we’ve got quite a lot of things that would be useful alternatives to the course that could be delivered locally”. IUCCP 7.

In addition, the pharmacists interviewed highlighted the lack of accreditation on the course and how it mainly focused on the telephone consultation rather than working on face-to-face skills with patients in the urgent care setting.

“What I was surprised at was when I went to the first day of the course there were two things that they said that I wasn’t impressed with. I know it was a bit out of their control but they said that the course isn’t accredited so you aren’t going to get any points at the end of it which to me doesn’t make sense. And also it seems that the course is just focusing on telephone consultation skills. It doesn’t really fit what our role is. 20 percent of our role is going to be doing the telephone skills but 80 percent isn’t. So I think it would be better if the course focused also on the face to face consultation not just the telephone skills”. IUCCP 6.

A number of pharmacists indicated the lack of feedback from the course on progress made and module completion.

“So the course. I just think I have learnt stuff from it, I do think it is kind of useful in some aspects, but I just don’t think it is clear enough, what we need to do and what we need to hand in. I have handed in the first module but I still don’t know if I have done the right thing, I handed it in last week was the deadline date and I’ve done the first module but I don’t know if I’ve done it right, and I’m kind of just waiting for feedback for someone to tell me you’ve done it completely wrong. But I know … Who I think is running that I think she has mixed up a few days and the problem is they are running the course, I think there is new one running in March and stuff like that, I think they are getting mixed up by who is doing what and who is on what cohort”. IUCCP 4.

All the IUCCPs commented that training to deal with mental health issues during 111 calls would be especially useful, particularly in the management of suicidal patients. Being an independent prescriber and being able to prescribe would also bring additional benefits to the service. (Focus group).

## 4. Discussion

The IUCCP programme is an innovative NHSE initiative. It provides an opportunity to extend the role of clinical pharmacists into the hard-pressed clinical environment of urgent and emergency care. The NENC initiative is a flagship programme and sets out to combine two components of urgent care. Its aim is to develop clinical pharmacists as practitioners working in the NHS 111 telephone service, combined with face-to-face experience in emergency departments and urgent care settings. This approach is felt to have advantages as it ensures professional accountability is maintained whilst clinical skills are consolidated with the whole emergency pathway experience.

Our evaluation highlighted four main themes, including Personality Traits, Integration, Benefits and Training. The Personality Traits theme highlighted enthusiasm, keenness and love of pressured situations that encouraged the pharmacists to apply and thrive in an urgent care setting. We would suggest that this might provide an opportunity to develop recruitment tools that aim to identify those clinicians most likely to thrive in the urgent care setting.

It is clear from the interview data that integration varied from Trust to Trust and for each individual, and many factors played a part, from lack of clarity during recruitment and definition of roles, lack of awareness of the benefits that IUCCPs could bring and lack of a structured induction. Although the Trusts and pharmacists have individual and varying perceptions of the benefits that pharmacists can bring to the urgent care setting, the integration process for some was straight forward and easy whilst others still struggle daily to find a clear role in their Trust setting. Despite this, it should be acknowledged that the programme is continuing to run with many success stories shared daily.

The increased presence of pharmacists with this developed perspective will be invaluable to services in which they are deployed, helping to further the ethos of integration across the system. It is believed that this innovative proposal is consistent with the ICS priority of whole-system working, and will be an exemplar of best practice.

A key theme emerged which highlights that training is key to progress in this programme, and NHSE has emphasised the importance of connectivity between clinical supervision and educational supervision, mentorship and learning sets.

The programme aims to further support the role of the pharmacist using decision support tools and so allow the pharmacists take a lead in the delivery of medicines/toxicology and pharmacy-related activities across the multidisciplinary IUC CAS team.

The longer-term aspiration is that IUC pharmacists will also able to work across organisational boundaries, providing enhanced support and resilience in the pharmacist shifts, as well as providing wider exposure to different clinical settings and the region’s urgent care system. In addition, North East Ambulance Service (NEAS) will develop remote working opportunities to enable pharmacists to undertake CAS shifts from their hospital base. It is worth noting that NEAS have been contracted to deliver the 111 (CAS) service since 1 October 2018. The development of this service is being implemented during a period of change, bringing its own challenges and pressures.

There are clear and obvious benefits of this program in this professional clinical workforce, but in order to ensure this is successful our study suggests that work should be initiated on convincing other parts of the system that the value that clinical pharmacists can bring is appropriate. Furthermore, it is vital that the pharmacists themselves, and their professional bodies, believe and can articulate their potential value and role.

Innovative pilots of this kind are of potential value, but their likelihood of success is not optimised without winning hearts and minds before implementation. The IUC pharmacists recruited in the North East of England were clearly highly motivated, enthusiastic clinical professionals who had attributes that made them ideally placed to work in this environment. However, they struggled to find their niche, and it has taken more time than is ideal in a time-limited pilot to integrate themselves into their teams. This is true of both the urgent care settings within their host Foundation Trusts and in the NEAS-hosted 111 service. Our study highlights the need, before developing innovative programmes, to ensure that those impacted by the role are fully aware of the purpose of the programme and its intended outcomes, and that teams embrace the concept and are welcoming of the IUCCPs in order to allow them to fully integrate as quickly as possible.

Notwithstanding these initial issues, our IUC clinical pharmacists have settled into their two new roles and are clearly thriving. They have clear career goals, aspirations and expectations, and will no doubt be an asset to our urgent care network. One of the outcomes from our interviews is the benefits that role models can have for those moving into new posts. Our hope is that the IUC clinical pharmacists know that they are becoming established and their value recognised, and that they will be the role models for future appointees.

Another additional benefit of the programme has been in the support of the transformation plan outlined in the Five-Year Forward View and the new Pharmacy Integration Fund. This has been accomplished with the deployment of clinical pharmacists and clinical pharmacy services in primary care, including urgent care settings such as NHS 111. This new service allows the development of “infrastructure” through the pharmacy professional workforce, accelerating digital integration and establishing the principles of medicines optimisation for patient-centred care.

## 5. Strengths and Limitations

The use of social constructionism as an underpinning theoretical perspective allowed an understanding that multiple realities exist [5] and to appreciate that individuals construct knowledge through their lives and interactions with others. This means that the information produced is reflective of a specific version of reality for these pharmacists. An additional strength of the study is the methodological approach. The use of framework analysis enabled this research to explore lived experiences, behaviours, emotions and feelings, as well as social movements [9]. The primary concern of the framework analysis is to describe and interpret what is happening in a specific setting, and therefore it has the greater potential to allow a deeper understanding of the issue being addressed [8].

A purposeful sample has its limitations, as interviewees were recruited from a small number of urgent care pharmacists. At the time of interviewing, eight urgent care pharmacists were employed, and all eight were recruited for interviewing. All the pharmacists interviewed were between the ages of 25–31, highlighting the limitations in term of disproportionally younger pharmacists recruited and interviewed.

A strength of this study is the use of semi-structured interviews, which enabled in-depth and detailed information regarding the integration of pharmacists into urgent care.

A reflective diary was used to review thoughts, feelings and experiences whilst interviewing and interpreting data. Although the use of a reflective diary does not eradicate bias from this study, it does however acknowledge the thought processes of the researcher and allowed the researcher to be more mindful of their own knowledge, especially during analytical procedures.

## 6. Future Research and Conclusions

From this study, additional benefits of integrating urgent care clinical pharmacists into urgent care settings have been highlighted, and in order to progress these benefits, we endorse the following. A formal and standardised induction to 111 services, development of standard operating procedures and guidelines for all pharmacists within 111 services be introduced. It is also imperative that each Trust develops multi-level awareness of the pharmacist’s role to prevent miscommunication, and finally that an advocate for this innovative programme within emergency and urgent care would be a huge advantage for the pharmacists and departmental integration.

In addition, further studies are recommended to investigate the advantages to patients, clinicians and NHS Trusts for the integration of urgent care clinical pharmacists into urgent care settings.

## Figures and Tables

**Table 1 pharmacy-07-00114-t001:** Interview schedule for 1:1 interviews.

1. Tell me a little bit about your career as a pharmacist?
2. What personally interested you in becoming a pharmacist in an urgent care setting?
3. Can you describe to me the split in time/roles within your new post?
4. Can you describe to me your role within each service?
5. What does a normal week look like for you?
6. Can you describe to me the process of deployment in *each setting? What worked well? What did not work so well? Did you feel colleagues understood your role? Were there any surprises? Is there anything that you think could be improved?
7. If this process of integration were to be repeated would you change anything? Could you give any advice for a new colleague following in your footsteps?
8. Can you describe the benefits (for you, patients and service) for placing you in this setting?
9. Do you think that integrating pharmacists into IUC is worthwhile and why?

* This was individulised in each interview.

**Table 2 pharmacy-07-00114-t002:** Four higher-order themes.

1. Personality Traits
2. Integration
3. Benefits
4. Training

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
