# Peer review of "Clinical Care Pharmacists in Urgent Care in North East England: A Qualitative Study of Experiences after Implementation"

_pharmacy, 2019, doi:10.3390/pharmacy7030114_

Round 1

Reviewer 1 Report

Thank-you for the opportunity to review this manuscript. I found the topic quite interesting, and the findings potentially useful for integrating pharmacists into novel settings such as urgent care. However, the manuscript needs much revision. The primary issues I have described below.

Title – needs changing to be clearer and less confusing. Suggest along the lines of ‘Clinical Care Pharmacists in Urgent Care in North East England: A Qualitative Study of Experiences after Implementation’

Abstract –

Introduction – Fairly well written, however a clearer preamble is needed especially for readers outside of the UK. I suggest describing the pharmacists’ current role in the UK and the need for their expanded practice and services.

Methods – The methods are fairly well described, though more detail is needed to allow replication of the study. Particularly relating to how the data was analysed, such as how many people analysed the data, how conflicting findings were settled, how interview data was triangulated with the focus group data, etc.

Results – Data on the characteristics of the participants is needed (age, gender, years of practice, level of education, etc.) to see if these experiences are possibly dependent of certain characteristics. I also suggest abbreviated ‘Urgent Care Pharmacist’

3.1 ‘Personality traits’ – the quotes used in this section are too lengthy and overshadow the actual finding. The finding needs more evidence and description in this section rather than a single sentence followed by massive participant quotes. Similar issues are present for 3.3 ‘Benefits’ and 3.4 ‘Training’ and (though less severe) for 3.2 ‘Integration’.

3.1 ‘The programme and its oversight’ appears to have been misplaced into the methods? I also don’t understand the purpose of this section (it is unclear on how it contributes to the manuscript)

Discussion – Most of the discussion is quite well done and doesn’t need any major changes apart from those made in the results that would then translate into the discussion. However a clear strengths and limitations section is needed, and discussion regarding future research that can further this field.

References – The references are appallingly done (both number and formatting), and require urgent attention by someone with experience in writing academic manuscripts.

Author Response

Thank you very much for your comments.

1-    Title – has been amended to ‘Clinical Care Pharmacists in Urgent Care in North East England: A Qualitative Study of Experiences after Implementation’

2-    Introduction –updates and now describes pharmacists’ current role in the UK and the need for their expanded practice and services.

3-    Methods – The methods are fairly well described, though more detail is needed to allow replication of the study. Particularly relating to how the data was analysed, such as how many people analysed the data, how conflicting findings were settled, how interview data was triangulated with the focus group data, etc.

4-     Results – Data on the characteristics of the participants age, gender, years of practice, level of education added

5-    The quotes used throughout results section reduced and more description and evidence added

6-    ‘The programme and its oversight’ section removed

7-    Strengths and limitation section added to the discussion including a future research section

8-    Referencing section fully reviewed and revised

Reviewer 2 Report

This is an interesting qualitative study using Grounded Theory in describing the embedding of clinical pharmacists within community urgent care and emergency settings sponsored through the NHS Foundation Trusts. The following comments and suggestions are given in order to highlight the manuscript's contributions to the literature and strengthen the overall presentation. 

Line 27 - there are no key words listed. I would suggest MeSH terms that would help others to find your manuscript in the vent that mdpi published your work.

Line 46 - the Integrated Urgent Care Pharmacist term is introduced. Provide an abbreviation in parenthesis (IUCP) there, and use the abbreviation through the manuscript.

Line 49 - How were the IUCPs selected? Was it a job requirement? Were incentives offered? IRB review? I would describe this aspect so as to address any coercion of participants that may affect their participation and/or statements.

Line 69 - please describe the sample in qualitative terms. It sounds like is was purposeful, since you had a purpose in selecting them. In addition, did you stop at eight IUCPs because you attained data saturation, i.e., that you stopped when you were not hearing anything new? Why did you stop with 8 participants?

Line 70 - consider changing "Eight semi structured interviews were conducted with the integrated urgent care pharmacists...." to "Semi structured interviews were conducted with eight IUCPs...."

Line 76 - please describe the method for fracturing the data. Was it a manual system of "cut and paste" or was a qualitative analysis software used, such as Atlas.ti or NVivo?

Line 110 - You uncovered 4 themes. How were the themes related to each other? There's a hint of this at lines 170-172 where you discuss the relationship between 'integration' and 'benefits.' 

Line 112 -  Beginning here, it would be helpful for the reader to have separation between the data quotes and the researchers' interpretation.

Line 161 - A&E is introduced. Please spell out the first time.

Line 372 - There is no conclusion section. However, table 3 seems to summarize statements made in the abstract.

References are incomplete and need to be formatted in mdpi style. While I realize that there is typically no literature search with grounded theory or qualitative studies in general, there may be helpful references that could place the study in the context of changes in NHS in general over the last decade..

Author Response

Thank you very much for your comments.

1-    Key words listed

2-    Use of abbreviated Integrated Urgent Care Clinical Pharmacist added (IUCCP)

3-    Section added on IUCCP’s section, job requirement and incentives

4-    More detail on purposive sampling added along with why we stopped at 8 participants

5-    Sentence structure changed as recommended "Eight semi structured interviews were conducted with the integrated urgent care pharmacists...." to "Semi structured interviews were conducted with eight IUCPs...."

6-    Method for fracturing data described

7-    As recommended, I have provided separation between the data quotes and the researchers' interpretation.

8-    As recommended, I have spelt out A and E the first time it is used

9-    Conclusion section added

Round 2

Reviewer 1 Report

Thank-you for updating the manuscript with the suggested revisions. I believe it is now acceptable for publication.

Reviewer 2 Report

The authors have addressed all the points raised adequately, and have included additional analysis of the meanings embedded within various themes. This inclusion makes the manuscript very readable and interesting. Reference style still needs scrubbed. Otherwise, well done!